# Effectiveness and Safety of Inactivated SARS-CoV-2 Vaccine (BBIBP-CorV) among Healthcare Workers: A Seven-Month Follow-Up Study at Fifteen Central Hospitals

**DOI:** 10.3390/vaccines11050892

**Published:** 2023-04-24

**Authors:** Rasha Ashmawy, Ehab Kamal, Wagdy Amin, Sandy Sharaf, Samar Kabeel, Raed Albiheyri, Yousra A. El-Maradny, Ebtisam Hassanin, Noura Elsaka, Ola Fahmy, Ahmed Awd, Heba Aboeldahab, Mai Nayle, Magda Afifi, Marwa Ibrahim, Raghda Rafaat, Shahinda Aly, Elrashdy M. Redwan

**Affiliations:** 1Clinical Research Department, Maamora Chest Hospital, MoHP, Alexandria 21923, Egypt; 2Infectious Diseases Administration, Directorate of Health Affairs, MoHP, Alexandria 21554, Egypt; 3Medical Research Division, National Research Center, Giza 12622, Egypt; 4General Administration of Chest Diseases, MoHP, Cairo 11516, Egypt; 5Clinical Research Department, Directorate of Health Affairs, MoHP, Damietta 34711, Egypt; 6Biological Science Department, Faculty of Science, King Abdulaziz University, Jeddah 21589, Saudi Arabia; 7Centre of Excellence in Bio Nanoscience Research, King Abdulaziz University, Jeddah 21589, Saudi Arabia; 8Protein Research Department, Genetic Engineering and Biotechnology Research Institute, City of Scientific Research and Technological Applications (SRTA-City), Alexandria 21934, Egypt; 9Microbiology and Immunology, Faculty of Pharmacy, Arab Academy for Science, Technology and Maritime Transport (AASTMT), Alamein 51718, Egypt; 10Clinical Pathology Department, Faculty of Medicine, New Valley University, New Valley 72713, Egypt; 11Clinical Research Department, Directorate of Health Affairs, MoHP, Sharkia 71529, Egypt; 12Egyptian Drug Authority, Alexandria 21532, Egypt; 13Physical Therapy Department, Kafr El-Sheikh General Hospital, MoHP, Kafr El-Sheikh 33511, Egypt; 14Clinical Research Department, Kom El-Shokafa Chest Hospital, MoHP, Alexandria 21572, Egypt; 15Clinical Research Department, Kafr El-Sheikh Chest Hospital, MoHP, Kafr El-Sheikh 33511, Egypt; 16Clinical Research Department, Fakous Central Hospital, MoHP, Sharkia 71529, Egypt

**Keywords:** COVID-19, vaccine effectiveness, healthcare workers, SARS-CoV-2, vaccination, workdays

## Abstract

Background: During a pandemic, healthcare workers are at high risk of contracting COVID-19. To protect these important individuals, it is highly recommended that they receive the COVID-19 vaccine. Our study focused on evaluating the safety and efficacy of Egypt’s first approved vaccine, the Sinopharm vaccine (BBIBP-CorV), and comparing these findings with other vaccines. Methods: An observational study was conducted in fifteen triage and isolation hospitals, from the 1st of March until the end of September 2021. The study included fully vaccinated and unvaccinated participants, and we measured vaccine effectiveness (using 1-aHR), the incidence rate of severely to critically ill hospitalized cases, COVID-19-related work absenteeism, and the safety of the vaccine as outcomes. Results: Of the 1364 healthcare workers who were interviewed, 1228 agreed to participate. After taking the hazard ratio into account, the vaccine effectiveness was found to be 67% (95% CI, 80–43%) for symptomatic PCR-confirmed cases. The incidence rate ratio for hospitalization was 0.45 (95% CI, 0.15–1.31) in the vaccinated group compared to the unvaccinated group, and there was a significant reduction in absenteeism among the vaccinated group (*p* < 0.007). Most adverse events were mild and well tolerated. Vaccinated pregnant and lactating mothers did not experience any sentinel adverse events. Conclusion: Our study found that the BBIBP-CorV vaccine was effective in protecting healthcare workers from COVID-19.

## 1. Introduction

In March 2020, the World Health Organization (WHO) announced that the novel coronavirus (SARS-CoV-2)-associated disease (COVID-19) was a pandemic that raised global concerns [1]. This was due to the rapid spread of the disease and the incredible increase in the number of cases and deaths due to SARS-CoV-2 respiratory symptoms and multi-organ damage, in addition to septic shock, as the number of cases reached 323 million cases with 5.5 million confirmed deaths globally until January 2022 [2]. Regarding Egypt, at the beginning of March 2021, the number of daily new cases was 586, and by the end of September, the number of cases was 741 per day [3]. Since the beginning of the illness, the medical personnel in hospitals around the world have taken on the responsibility of protecting and combating this dreadful disease for the benefit of all humanity. Consequently, numerous healthcare professionals (HCWs) have contracted COVID-19 and become victims of the disease. The World Health Organization (WHO) has approximated that between January 2020 and May 2021, the fatalities of HCWs due to COVID-19 were in the range of 80,000 to 180,000 [4]. Therefore, the Centers for Disease Prevention and Control (CDC) recommended the vaccination of HCWs to prevent morbidity and mortality among global frontline defenders [5]. Moreover, HCWs are exposed daily to a remarkably high viral load from their work environment, which leads to increased susceptibility to becoming sick easily and the loss of workdays, which has a critical influence during the pandemic. In addition, HCWs can spread the infection to their families, which highlights the need for vaccinating healthcare workers [6].

Regrettably, there has been contradictory information regarding the most secure and efficient treatment for COVID-19, thus prompting numerous pharmaceutical companies to strive for developing a SARS-CoV-2 vaccine since the outbreak. As of 24 January 2022, 26 vaccines have been granted emergency use authorization or authorized for usage outside clinical trials. Egypt has approved nine vaccines for use, with four of them still undergoing clinical trials [7]. The Egyptian Ministry of Health and Population (MoHP) collaborated with Sinopharm Pharmaceutical Company during the phase 3 trial of the vaccine. Upon conducting an interim analysis, it was determined that the BBIBP-CorV vaccine was safe, effective, and had suitable storage conditions [8]. This led to the government acquiring successive batches of the inactivated vaccine in January 2021, which became the first type of vaccine to arrive in Egypt. The initial phase of the vaccination campaign began by administering the vaccine to HCWs who encounter COVID-19 patients regularly in the chest, fever (triage), and isolation hospitals.

As a result of the urgent demand for vaccination, there is currently inadequate long-term real-life follow-up data available [9]. Moreover, there is a lack of comprehensive studies on the efficacy of the BBIBP-CorV vaccine in this crucial group. Therefore, our objective was to evaluate the safety and effectiveness of this vaccine in the daily lives of legitimate healthcare workers. Furthermore, our goal was to compare these outcomes with the findings of other studies conducted on COVID-19 vaccination among the same population.

## 2. Methodology

### 2.1. Study Design and Settings: Participants

An observational study was carried out in fifteen hospitals that operated as isolation and triage facilities during the study period. These hospitals and their staff were deemed to be at a high risk of COVID-19 infection due to the substantial viral load present in these settings.

### 2.2. Materials and Methods

We performed a multistage sampling technique. First, we divided Egypt into five main clusters and chose the governorate with the greatest population according to the latest census of Egypt (CAMPAS 2020): Middle Egypt (Cairo governorate), East Delta (El Sharika governorate), West Delta (Alexandria governorate), North Delta (Kafr El-Sheikh governorate), and South Delta (Assuit governorate). Then, we stratified hospitals by specialty to randomly select one isolation, one fever (infectious diseases hospital), and one chest hospital from each governorate to participate. Hence, we included five isolations, five fevers, and five chest hospitals scattered across Egypt.

The study began after the ethical approval from the MoHP Research Ethics Committee (Com.No/Dec.No:10-2021/10). The site investigators were chosen from each hospital to prepare a sample frame from the staff of their hospitals. Then, the site investigators randomly selected HCWs (physicians, nurses, pharmacists, dentists, and technicians) from the sample frame to perform the first interview and determine whether they had been vaccinated with the inactivated BBIBP-CorV vaccine or not. The study included healthcare workers who had received two doses of the inactivated BBIBP-CorV vaccine, with a 21-day interval between the doses. Partially vaccinated individuals were not encountered during the study. The control group comprised healthcare workers who either did not receive the vaccine or were unwilling to receive it during the research period. We excluded any HCWs who were vaccinated with another type of SARS-CoV-2 vaccine or who refused to participate in the first interview. Unfortunately, due to the low number of healthcare workers who received the BBIBP-CorV vaccine during this time frame, the researchers were compelled to include nearly all individuals who had been vaccinated with BBIBP-CorV.

The follow-up period spanned nearly seven months, commencing 14 days after the administration of the second vaccine dose for each vaccinated healthcare worker and concluding on September 30th, 2021. In the case of the unvaccinated group, we set March 1st as the start of the follow-up period by estimating the median time for initiating follow-up in the vaccinated group, as vaccination times varied. We were unable to continue the study beyond September 30th, because vaccination became mandatory, resulting in the loss of the control group.

### 2.3. Sample Size and Calculation

From the literature, nearly 10% of COVID-19-confirmed cases are severely to critically ill (our primary outcome) [10,11], and we assumed that the vaccine would reduce this percentage by 50%. In addition, by using Stata 16 software for sample size calculation, a power of 80%, and a 0.05 level of significance, the minimum total accepted sample size was 950 participants. Upon adjusting the cluster effect using the varying cluster size equation, the total sample size was multiplied with inverse minRE = 1.18, assuming the coefficient of variation (CV = 0.6) and the minimum relative efficiency (minRE = 0.847) [12]. Consequently, 1125 was the adjusted sample size.

### 2.4. Data Source and Measurements

During the initial interview with the participating healthcare workers, two distinct Google forms were employed. The first form was utilized to gather baseline data and any outcomes that had arisen before the interview, while the second form was employed to monitor any adverse events that occurred during the week following each vaccine dose. The baseline data collected during the initial interview included demographics, profession, workplace, daily contact status with COVID-19 patients, comorbidities, risk factors for severe COVID-19 infection, pregnancy, lactation, smoking status, prior COVID-19 infection, lost workdays due to COVID-19 illness, and influenza vaccination status from the previous year. Additionally, the dates of the first and second doses of the BBIBP-CorV vaccine were obtained from hospital registries. Furthermore, all participants were asked about any confirmed or suspected COVID-19 infection, which was retrospectively started 14 days after full vaccination, or from the 1st of March for the unvaccinated group. The HCWs who received the BBIBP-CorV vaccine were also queried about any solicited adverse events they experienced after vaccination, including the severity of the events, their frequency, duration, and the treatment utilized [13]. After September 2021, the dates of any study outcomes that occurred were obtained, either by re-interviewing or calling the participants or from the hospital’s data for individuals who were unable to be reached due to death or critical illness.

### 2.5. Outcomes

The primary outcomes were assessing three separate endpoints: vaccine effectiveness to prevent a PCR-confirmed symptomatic COVID-19 infection; severity outcome was the incidence rate of hospitalized cases (severely to critically ill), or economic outcomes by comparing the median workdays lost due to COVID-19 infection between the two groups. The secondary outcomes were: (1) The incidence rate of total COVID-19 infection either PCR-confirmed or -suspected, as defined in the MoHP protocol [14]; the suspected case was any HCW that had three or more typical COVID-19 symptoms: fever, cough, anosmia, ageusia general weakness, headache, myalgia, coryza, dyspnea, nausea/vomiting, diarrhea, or altered mental status. (2) Mechanical ventilation (MV) incidence rate and the COVID-19 mortality rate. (3) The numbers and proportion of solicited adverse events after vaccination either local or systemic and the reporting of any serious adverse events in the vaccinated group, where local solicited adverse events were pain, erythema, swelling, or lymphadenopathy at the site of injection. In addition to systematic adverse events, there were rash, fever, chills, fatigue, myalgia, arthralgia, nausea, vomiting, change in blood pressure, or headache.

### 2.6. Statistical Analysis and Statistical Package

Data were analyzed using R software 4.1.1. packages, with a 5% level of significance. The median and interquartile ranges were used to describe numerical data. Number and percentage were used for categorical data. All baseline data: demographics, comorbidities, risk factors, previous influenza vaccination, and prior COVID-19 infection were compared in both groups by using the chi-square test or Mann–Whitney test, and variables with statistically significant differences between the two groups were adjusted either in subgroup analysis or in multivariate Cox-regression. The chi-square test and incidence rate ratio (IRR) were used for comparing the significance of outcomes (PCR-confirmed symptomatic case, total COVID-19 infection, hospitalization, mechanical ventilation, and death). A subgroup analysis was performed for IRR. Vaccine effectiveness was calculated as 1-adjusted hazard ratio (HR) through Cox-Hazard regression. In addition to the number and percentage of the solicited adverse events reported seven days after each dose, they were then classified by severity according to the US FDA grading [15].

### 2.7. Literature Review

There are several studies conducted on HCWs to evaluate the safety and effectiveness of vaccines protecting from COVID-19 infection. We conducted aPubMed search using the keywords “Vaccine effectiveness in health care workers” to compare the results with our study.

## 3. Results

From 1364 interviewed HCWs, 1228 were enrolled in the study (Figure 1), the median follow-up days was 213 (IQR, 180–213), and the unvaccinated persons were younger than those vaccinated, with median ages of 30 (IQR, 26–37) and 35 (IQR 29–42) years, respectively. The female sex was more predominant in the unvaccinated group; two females were pregnant (they got vaccinated accidentally in early pregnancy) and one was a lactating mother in the vaccinated group, while 19 were pregnant and 26 were lactating in the unvaccinated group. Nurses were the highest category among the unvaccinated group while pharmacists were in the vaccinated one. A total of 57.3% of the unvaccinated group worked in chest hospitals and 43.0% of the vaccinated group worked in isolation (Appendix A). The majority of both groups did not have any risk factors for severe COVID-19 infection and were not smokers. There was no statistically significant difference in both groups for prior COVID-19 infection, 16.7% for unvaccinated, and 20.3% for vaccinated (*p* = 0.207), in addition to the loss of workdays due to prior COVID-19 infection (*p* = 0.142). All the baseline characteristics of the first interview are discussed in (Table 1 and Appendix A).

### 3.1. Effectiveness Estimation against COVID-19 Infection

During the study follow-up period that ended in September 2021, the proportion of PCR-confirmed cases was statistically significant between unvaccinated and vaccinated groups, respectively: 14.4% and 5.5% (*p* < 0.001). There is a statically significant difference between the 2 groups in the number of workdays lost due to COVID-19 infection as being greater in the unvaccinated group (median 14, IQR 10–15) than in the vaccinated group (10 days, IQR 6.5–15) days (Table 2).

Among 25 hospitalized unvaccinated HCWs, 4 were admitted to the ICU and mechanically ventilated (MV), whereas one was MV from 4 hospitalized vaccinated HCWs. However, the hospitalization and MV by COVID-19 infection between the two groups were statistically insignificant, (*p* = 0.108) and (*p* = 0.728), respectively, and only one death occurred among the unvaccinated ones. Kaplan–Meier curves showed significance for obtaining PCR-confirmed COVID-19 cases and total COVID-19 cases, *p* < 0.0001 and 0.006, respectively (Appendix A), where there was no significant difference between hospitalization and MV (Appendix A). Then, vaccine effectiveness was calculated by conducting a multivariate Cox regression analysis of 67% (95% CI, 80–43%) to protect against symptomatic PCR-confirmed cases and 46% (95% CI, 62–24%) for total COVID-19 infection after adjusting HR for age, gender, hospital type, contact to COVID-19 patients, workplace, prior COVID-19 infection, and influenza vaccination. Crude incidence rate ratios (IRRs) of the vaccinated to unvaccinated group were 0.39 (95% CI, 0.24–0.65) for PCR-confirmed symptomatic cases, 0.63 (95% CI, 0.46–0.86) for total COVID-19 infection, 0.45 (95% CI, 0.15–1.31) for hospitalization, 0.71 (95% CI, 0.07–6.39) for MV, and could not be calculated for death (Table 2).

### 3.2. Adjusting Variables

Upon subgrouping of the incidence rate ratio (IRR) for PCR-confirmed cases of vaccinated to unvaccinated, statistically significant variables were age less than 50 (IRR = 0.38, 95% CI 0.23–0.65), direct contact to COVID-19 patients (IRR = 0.2, 95% CI 0.09–0.43), chest hospitals (IRR = 0.31, 95% CI 0.12–0.78), and no prior COVID-19 infection (IRR = 0.4, 95% CI 0.24–0.67). Another subgrouping of the incidence rate ratio (IRR) for total COVID-19 and hospitalized COVID-19 cases is shown in Appendix A.

### 3.3. Safety Measures

Mostly, the adverse events (AEs) were mild (grade 1). Regarding local adverse events (AEs) reported: Half of the vaccinated suffered from local pain (the most prevalent local AEs), contributing to 52.5% following both the 1st and the 2nd dose, and swelling appeared around 3.3% following the 1st dose. No local pain associated with erythema and swelling, numbness, or lymphadenopathy appeared following the administration of the 2nd dose only. Systemic AEs reported the following: fever mainly followed the first dose for 34.2% (temperature range, 37.3–39 °C), the largest duration was for 3 days, and the fluctuation in blood pressure appeared following the administration of both doses, as follows: 6 vaccinated HCWs suffered from an increase in their blood pressure (150/90 mmHg was the highest among them), while there was a decrease in blood pressure in one vaccinated (80/50 mmHg) with a maximum duration of 3 days; other AEs and their severity are shown in Figure 2. Besides this, unsolicited AEs were reported as follows: mild edema occurred in one vaccinee in the face and hands following the administration of the first vaccine dose, and hyperpigmentation was reported from one vaccinee and lasted for 4 months; pregnant and lactating mothers did not report any sentinel AEs (Appendix A). Treatments for these adverse effects were ACEIs, CCBs, and BB for increased blood pressure; Paracetamol for pain or fever was the most commonly used drug for adverse events management followed by other NSAIDs; antihistaminic injection was for edema.

### 3.4. Literature Review

There are several studies conducted on HCWs to evaluate the safety and effectiveness of vaccines protecting from COVID-19 infection. Studying different research papers on PubMed using the keywords “Vaccine effectiveness in health care workers” yields 20 studies from various countries, as presented in Table 3. Different types of vaccines were used, including those from Pfizer-BioNTech, Moderna, Janssen, Oxford-AstraZeneca, and other inactivated vaccines. The VE of these vaccines in HCWs ranged from 66% to 99%. Many factors were documented to increase the vaccine efficacy: 1. the administration of a second and booster dose significantly increased vaccine efficacy [16,17]; 2. the interval time increased between vaccine doses [18]; 3. Moderna had the highest documented vaccine effectiveness [19]; 4. SARS-CoV-2 variants of concern (VOCs) that were significantly affecting VE were Delta and Omicron, which decreased VE [20,21].

## 4. Discussion

The COVID-19 pandemic poses a significant risk to healthcare workers, making them more vulnerable to contracting the virus. In this observational study, we analyzed infection rates of confirmed COVID-19 among a large cohort of Egyptian healthcare workers (n = 1228) over a 7-month follow-up period from March 2021 to September 2021. Out of the vaccinated healthcare workers, we identified 18 cases of COVID-19, which represents 5.5% of the cohort. The incidence rate ratio was 0.39 (95% CI, 0.24–0.65), indicating a significantly lower incidence rate of COVID-19 among the vaccinated group. We also found that the inactivated BBIBP-CorV vaccine had an effectiveness of 67% (95% CI 80–43%) among healthcare workers, calculated as a percentage of (1-adjusted HR). Furthermore, we are concerned about the number of workdays missed by healthcare workers, as they play a vital role in combating disease within the population. Their absence could lead to a shortage of COVID-19 treatment services.

The Emergency authorization handed out the inactivated whole virion vaccine (BBIBP-CorV); it was the initial launch of a vaccination program against COVID-19 for HCWs that began on 16 January 2021. The BBIBP-CorV vaccine was not as effective against variants of SARS-CoV-2 as the original virus [37]. However, our timeline study could ensure its effectiveness against the Delta variant, with the second surge linked to the highly transmissible Delta variant. On the 28th of July 2021, it appeared in 132 countries globally. Moreover, the WHO Eastern Mediterranean Region Office reported the detection of Delta variants in 15 countries, including Egypt [38]. The Egyptian Ministers of Health announced the first Delta variant case in Egypt in July 2021 [39]. This variant is the predominant one now in Egypt [40]. Consequently, the first detection of this variant in India in March 2021 [36] did not block its appearance before in Egypt. Hence, our targeted population in their hospitals deals with all COVID-19 patients, and they may be infected by existing variants during the follow-up period.

Moreover, there was a statically significant difference between the two groups in the number of workdays lost due to COVID-19 infection; the days lost were greater in the unvaccinated group, which will certainly impact the provision of health services and consequently be economically effective. In addition, the vaccine effectiveness (VE) for hospitalization or mechanical ventilation due to COVID-19 infection was insignificant between the two arms, which may be due to the good planning and innovation in public health to control the infection outbreak, Egyptian protocol management, less-severe new variants of the virus, and our median age groups.

The VE in our study is similar to estimates that have been reported in New Delhi, India, for the SARS-CoV-2 reinfection rate and estimated effectiveness of the inactivated whole virion vaccine BBV152 among healthcare workers, and fully vaccinated HCWs had a lower risk of reinfection (HR = 0.14, 95% CI 0.08–0.23) [41]. Despite their higher VE, which may be due to their larger sample size or different vaccine type, their study was conducted at only one large hospital. In addition, a retrospective cohort study with 176,640 individuals was conducted in the UAE to assess the BBIBP-CorV vaccine’s efficacy in the real world. The outcomes are consistent with our trial, in which the vaccine successfully averted hospitalization, admissions for critical care, and death. Participants who received all three doses of the vaccine had corresponding vaccine efficacy rates of 80%, 92%, and 97% [42].

Another study comparing the BBIBP-CorV and the mRNA BNT162b2 (Pfizer-BioNTech) vaccination in the UAE was performed concerning the Delta (B.1.617.2) variant. In comparison to Pfizer-mRNA BioNTech’s BNT162b2, BBIBP-CorV showed 95% (95% CI: 94, 97%) protection against COVID-19-related hospitalization from the Delta (B.1.617.2) variant [43]. Additionally, a case–control study was carried out in Morocco to ascertain the actual BBIBP-CorV VE against the serious or critical hospitalization of people who tested positive for COVID-19. The study showed that BBIBP-CorV offers greater protection to working-age adults compared to older adults. In individuals over the age of 18, the adjusted protective effectiveness of two doses of the vaccine varied from 93.9% to 100%, but in adults 60 and older, it was only 53.3% [44].

The first dose of the BBIBP-CorV vaccine was 89% effective against hospitalization brought on by critical COVID-19 infection, according to a different real-world trial carried out in Morocco; older persons showed reduced efficiency in comparison to younger adults. Although the vaccine’s two-dose VE was 96% effective against critical COVID-19 hospitalization, it was only 53% effective in older individuals. Compared to men, women had a somewhat higher adjusted VE. No one who received the booster dose needed to be hospitalized, due to a serious COVID-19 infection, regardless of age.

Two doses of the inactivated BBIBP-CorV vaccine were found to be, respectively, 79.6%, 86%, and 84.1% effective in preventing hospitalization for COVID-19, critical care admission, and mortality in a different study conducted in Abu Dhabi with a 12-month follow-up. This study also noted that the age-related decline in vaccine protection against hospitalization was supported by a lower level of vaccine protection in the older age group (>60 years) compared to the younger age group [45]. This study found that there was no significant difference in mortality between males and females when it came to how well the vaccine prevented hospitalization (82.3%) compared to 74.5% in males.

Until now, the available data suggest that effectiveness is highest in RNA vaccines than in viral vector vaccines and lowest in the inactivated SARS-CoV-2 vaccines. However, Israel recently reported a breakthrough infection of SARS-CoV-2, dominated by variant B.1.1.7 in a small number of fully vaccinated healthcare workers, raising concerns about the effectiveness of the original vaccine against those variants [46], whereas another study in Egypt showed that the BBIBP-CorV vaccine cannot trigger sufficient antibodies as an immune response in most vaccinated cases [37]. The UAE solved this problem by administering a third booster dose of the BBIBP-CorV vaccine to improve the antibody response. Subsequently, more studies are needed on those who receive three doses of the BBIBP-CorV vaccine to determine the vaccine’s effectiveness among them. On the other hand, we adjusted for previous COVID-19 infections 14 days after being fully vaccinated in the vaccinated group and before March 1st for the unvaccinated group, to encompass high titers of immune antibodies, and we did not find any difference. Indeed, another study examined the effectiveness of one of the COVID-19 vaccines with the influenza vaccine to assess immunogenicity [47]. So, we adjusted this variable in our study, and there was no difference between the two groups.

The vaccine works by using killed viral particles to expose the body’s immune system to the virus without risking a serious disease response [37]. In our study, the post-vaccine symptoms after the first and second doses are commonly mild, as classified by the participants. About half of the participants developed local mild reactions and local pain. Fever was also mild and was highest following the first dose (34.2%) with the highest temperature (39 °C) with a maximum of 3 days. Similar to the findings of an earlier study, it is noticeable that a higher percentage of participants did not feel any side effects after the second dose of vaccination than after the first dose [48], which may be attributed to the immune system response (increased cytokines). Other adverse events were variable and mostly occurred in one patient: mild edema in the face and hands, hypertension, and hypotension.

After all that, there is hesitancy in accepting vaccination among people, accepted mainly by young HCWs with fewer comorbidities and risk factors for admission. Our study supports the governmental efforts to improve vaccination coverage and to impose it on people in many sectors. Finally, vaccination against SARS-CoV-2 is the solution to overcome this disaster infection.

Strengths: Our study has at least three main strengths. First, it was conducted across multi-centers distributed all over Egypt; fifteen hospitals were included, chosen by the multistage sampling technique—the highest infection rate areas. Second, we targeted healthcare workers as they are the theater of war in this pandemic disease, exposed to high viral load. Third, we also handled a subgroup analysis for all variables for our outcomes.

Limitations: This study was observational, so the causal relationship between adverse events and vaccine administration could not be well defined. Passive collection of adverse events was conducted after vaccination. The study was not specifically intended to assess the efficacy of the vaccine against asymptomatic infection and may not have had sufficient power to determine the efficacy of the vaccine against MV and mortality. We need further studies for other categories of populations such as children, and pregnant and lactating women to check for the effectiveness and safety of this vaccine among them and with longer duration. In addition, we may miss some asymptomatic patients during the follow-up period.

## 5. Conclusions

The inactivated BBIBP-CorV (from Sinopharm) vaccine proved to have comparable effectiveness in avoiding and reducing hospitalizations, critical care admissions, and mortality linked to COVID-19 in healthcare providers.

## Figures and Tables

**Figure 1 vaccines-11-00892-f001:**
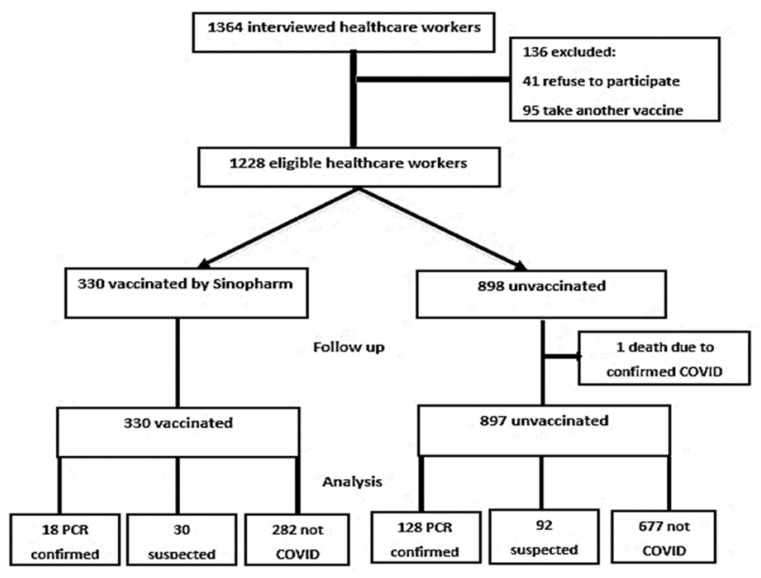
Consort diagram showing study participants.

**Figure 2 vaccines-11-00892-f002:**
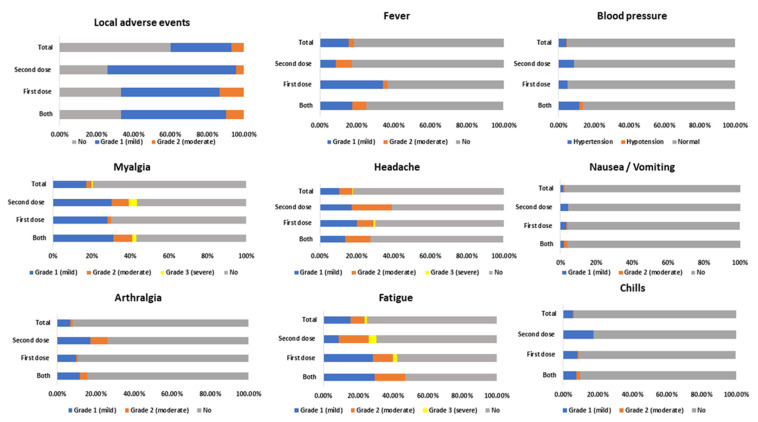
Solicited adverse events.

**Table 1 vaccines-11-00892-t001:** Baseline characteristics at the first interview.

Characteristics, No. (%)	Unvaccinated (N = 898)	Vaccinated (N = 330)	Total (N = 1228)	*p*-Value
Age category				0.003
<50 years	841 (93.9%)	295 (89.3%)	1136 (92.5%)	
≥50 years	55 (6.1%)	37 (11.2%)	92 (7.5%)	
Gender				<0.001
Female	777 (86.5%)	228 (69.1%)	1005 (81.8%)	
Male	121 (13.5%)	102 (30.9%)	223 (18.2%)	
Healthcare Profession				<0.001
Dentist	15 (1.7%)	1 (0.3%)	16 (1.3%)	
Nurse	569 (63.4%)	71 (21.5%)	640 (52.1%)	
Pharmacist	117 (13.0%)	132 (40.0%)	249 (20.3%)	
Physician	127 (14.1%)	79 (23.9%)	206 (16.8%)	
Technician	70 (7.8%)	47 (14.2%)	117 (9.5%)	
Contact status to COVID-19 patients				<0.001
Direct contact	728 (81.1%)	159 (48.2%)	887 (72.2%)	
Indirect contact	170 (18.9%)	171 (51.8%)	341 (27.8%)	
Influenza vaccination in 2020				<0.001
No	685 (76.3%)	193 (58.5%)	878 (71.5%)	
Yes	213 (23.7%)	137 (41.5%)	350 (28.5%)	
Number of risk factors for severe COVID-19				<0.001
0	718 (80.0%)	227 (68.8%)	945 (77.0%)	
1	135 (15.0%)	70 (21.2%)	205 (16.7%)	
≥2	45 (5.0%)	33 (10.0%)	78 (63%)	
Smoking status				0.108
No	865 (96.3%)	311 (94.2%)	1176 (95.8%)	
Yes	33 (3.7%)	19 (5.8%)	52 (4.2%)	
Pregnancy				0.07
No	758(97.6%)	226 (99.2%)	984 (97.9%)	
Yes	19 (2.4%)	2 (0.8%)	21 (2.1%)	
Lactation				0.006
No	751 (96.7%)	227(99.6%)	978 (97.3%)	
Yes	26 (3.3%)	1 (0.4%)	27 (2.7%)	
Previous COVID-19 infection before vaccination				0.207
No	748 (83.3%)	263 (79.7%)	1011 (82.3%)	
Yes	150 (16.7%)	67 (20.3%)	217 (17.6%)	
Previous hospitalization due to COVID-19				0.924
No	861 (95.9%)	316 (95.8%)	1177 (95.8%)	
Yes	37 (4.1%)	14 (4.2%)	51 (4.2%)	
Number of workdays lost due to previous COVID-19				0.142
Median (IQR)	14 (10–15)	14 (10–15)	14 (10–15)	
Total days lost	2034	1037	3071	

Risk factors for severe COVID-19: chronic lung disease, cardiovascular disease, diabetes, hypertension, BMI > 35, chronic kidney disease, chronic liver disease, and autoimmune diseases. Age > 50: According to the COVID-19 protocol, this group has a high risk of getting a severe infection. Tests of significance used between the 2 groups: Chi-square, Fischer Exact, and Mann–Whitney.

**Table 2 vaccines-11-00892-t002:** Outcomes at the end of September 2021.

	Unvaccinated (N = 898)	Vaccinated (N = 330)	Total (N = 1228)	*p*-Value	IRR (95% CI)	Unadjusted HR (*p*-Value)	Adjusted HR (95% CI)	VE (95% CI)
Current COVID-19 infection				<0.001				
No	677 (75.4%)	282 (85.5%)	959 (78.1%)		Reference	Reference	Reference	
PCR confirmed	129 (14.4%)	18 (5.5%)	147 (12.0%)		0.39 (0.24; 0.65)	0.42 (*p* = 0.0007)	0.33 ^a^ (0.2; 0.57)	67% (80; 43%)
Total COVID	221 (24.6%)	48 (14.5%)	269 (21.9%)		0.63 (0.46; 0.86)	0.64 (*p* = 0.004)	0.54 ^a^ (0.38; 0.76)	46% (62; 24%)
Hospitalization due to COVID-19				0.108				
No	873 (97.2%)	326 (98.8%)	1199 (97.6%)		Reference	Reference	Reference	
Yes	25 (2.8%)	4 (1.2%)	29 (2.4%)		0.45 (0.15–1.31)	0.48 (*p* = 0.14)	0.35 ^b^ (0.12–1.08)	65%
Admission place				0.326				
ICU	4 (0.4%)	1 (0.3%)	5 (0.4%)		----	----	----	
Intermediate care	2 (0.2%)	1 (0.3%)	3 (0.2%)		----	----	----	
Ward	19 (2.1%)	2 (0.6%)	21 (1.7%)		-----	----	----	
Mechanically ventilated				0.728				
No	894 (99.6%)	329 (99.7%)	1223 (99.6%)		Reference	Reference	Reference	
Yes	4 (0.4%)	1 (0.3%)	5 (0.4%)		0.71 (0.07; 6.39)	0.95 (*p* = 0.96)	1.26 ^c^ (0.13; 12.2)	--
Still alive				0.544				
No	1 (0.1%)	0 (0.0%)	1 (0.1%)		----	----	----	
Yes	897 (99.9%)	330 (100%)	1227 (99.9%)		-----	----	----	
Workdays lost due to current COVID-19				0.007				
Median (IQR)	14 (10–15)	10 (6.5–15)	14 (19–15)		-----	----	----	
Total	2824	557	3381		-----	----	----	

IRR: incidence rate ratio of vaccinated to unvaccinated per person-days, HR: crude hazard ratio vaccinated/unvaccinated HCWs, adjusted HR: adjusted hazard ratio, VE: vaccine effectiveness = (1—adjusted HR) %. ^a^ Adjusted for age, gender, contact to COVID-19 patients, hospital type, workplace, previous COVID-19 infection, and influenza vaccination. ^b^ Adjusted for age, gender, contact with COVID-19 patients, previous COVID-19 infection, number of risk factors, pregnancy, and smoking. ^c^ Adjusted for age and gender. Test of significance used between the 2 groups: Chi-square, Fischer Exact, and Mann–Whitney.

**Table 3 vaccines-11-00892-t003:** Vaccine effectiveness from different studies.

Author, Year	Country	Study Design	Population	Type of Vaccine	Vaccine Effectiveness Against SARS-CoV-2 Infection	Doses Received	Type of Variants
Fowlkes, 2021 [21]	USA	Cohort	4217 HCWs	Pfizer-BioNTech, Moderna, and Janssen COVID-19 vaccines	Before and during Delta variants, they were 80% and 66%, respectively.	2 doses	Before and during Delta (B.1.617.2)
Spitzer, 2022 [22]	Israel	Prospective cohort	1928 HCWs	BNT162b2 vaccine	After the booster dose, it was 97.7%.	3 doses	Before omicron
Özgür, 2022 [23]	Turkey	Prospective observational study	276 HCWs	Inactivated vaccine	Antibody levels after 1st and 2nd doses of the vaccine were 27.2% and 98.5%, respectively.	2 doses	Not specified
Thompson, 2021 [24]	USA	Randomized placebo-controlled Phase III trials	3950 HCWs	BNT162b2 or mRNA-1273 vaccine	After the 1st and 2nd doses of the vaccine, they were 80% and 90%, respectively.	≥1 dose	Not specified
Akar, 2021 [25]	Turkey	Prospective observational study	1053 HCWs	CoronaVac^®^ inactivated vaccine	The IgG amounts after first and second doses of the vaccine were 25.3% and 97.9%, respectively. The nAb positivity was 97.7% after the 2nd dose.	≥1 dose	Not specified
Pilishvili, 2021 [26]	USA	Test-negative case–control study	1482 HCWs	BNT162b2 and mRNA-1273 vaccine	After 1st and 2nd doses of the Pfizer vaccine, they were 77.6% and 88.8%, respectively. After the 1st and 2nd doses of the Moderna vaccine, they were 88.9% and 96.3%, respectively.	≥1 dose	Not specified
Herzberg, 2022 [27]	Germany	Longitudinal observational study	184 HCWs	BNT162b2 (Pfizer-BioNTech) vaccine	IgG response was 100%, and 53.11% after the 2nd dose and 9 months after the 2nd dose, respectively. nAb was 68.20% after 9 months of the 2nd dose.	2 doses	Not specified
Tré-Hardy, 2021 [28]	Belgium	Prospective cohort study	201 HCWs	mRNA-1273 (Moderna) vaccine	Significant decrease in the antibody response after 6 months from the 2nd dose of vaccine.	2 doses	Not specified
Arankalle, 2022 [29]	India	Observational cohort study	187 HCWs	COVISHIELD and COVAXIN	98% protection after COVISHIELD and 99% after COVAXIN; 95% protection after 6 months.	2 doses	Delta (B.1.617.2)
Kunno, 2022 [17]	Thailand	Cross-Sectional Study	780 HCWs, general population	Viral vector vaccine and inactivated vaccine	HCW more satisfied with 3 doses of vaccines.	≥2 doses	Not specified
Jiménez-Sepúlveda, 2022 [30]	Spain	Case–control study	1404 HCWs	BNT162b2 (Pfizer-BioNTech) vaccine	91.9%, 63.7%, and 37.2% protection after (12 to 120 days), (121 to 240 days), and (241 days) from the 2nd dose.	2 doses	Alpha (B.1.1.7) and Delta (B.1.617.2)
Kim, 2022 [31]	Korea	Prospective longitudinal Study	32 HCWs	2ChAdOx1 and booster dose of BNT162b2	The effectiveness of 2nd dose was 12.0%, and 50% after the booster dose.	3 doses	Delta and Omicron
Larese Filon, 2022 [32]	Italy	retrospective cohort study	4251 HCWs	BNT162b2 (Pfizer-BioNTech) vaccine	Vaccine effectiveness 95%.	2 doses	Delta (B.1.617.2)
Leong, 2023 [18]	Canada	A prospective cohort study	309 HCWs	BNT162b2 (Pfizer-BioNTech) vaccine	Increasing dose intervals significantly increased the vaccine efficacy.	2 doses	Original and Beta (B.1.351) variants
Evans, 2022 [20]	USA	Experimental study	48 HCWs	mRNA-1273 (Moderna) or BNT162b2 (Pfizer-BioNTech) vaccine	Neutralizing antibodies decreased after 6 months from the 2nd dose with 56.3% and 89.6%, against Delta and Omicron, respectively.	2 doses	D614G, Alpha (B.1.1.7), Beta (B.1.351), Delta (B.1.617.2), and Omicron (B.1.1.529),
Solis-Castro, 2022 [33]	Peru	Retrospective cohort study	520,733 HCWs	Inactivated Vaccine	90.9%, 67.7%, and 26.3% effectiveness for death, hospitalizations, and COVID-19 infection, respectively.	2 doses	Alpha (B.1.1.7), Beta (B.1.351), and Gamma (P1)
Angel, 2021 [16]	Israel	Retrospective cohort study	6710 HCWs	BNT162b2 (Pfizer-BioNTech) vaccine	Effectiveness was 97% symptomatic and 86% asymptomatic infection.	≥1 dose	During Alpha (B.1.1.7)
Lumley, 2022 [34]	United Kingdom	longitudinal cohort study	13,109 HCWs	BNT162b2 (Pfizer-BioNTech) or Oxford-AstraZeneca (ChAdOx1 nCOV-19) vaccine	Symptomatic infection 1st dose ↓67%, 2nd dose ↓90%. Asymptomatic infection 1st dose ↓64%, 2nd dose ↓85%.	≥1 dose	Alpha (B.1.1.7)
Vaishya, 2021 [35]	India	Multicenter cohort study	28,342 HCWs	Oxford-AstraZeneca (ChAdOx1 nCOV-19) or inactivated vaccine (BBV152)	Symptomatic infection AstraZeneca: ↓94.89% BBV152: ↓95.42%.	≥1 dose	Alpha (B.1.1.7), Delta (B.1.617.2)
Malhotra, 2022 [36]	India	Retrospective cohort study	15,244 HCWs	Inactivated whole virion vaccine (BBV152)	86% in preventing reinfection after 2nd dose.	≥1 dose	Delta (B.1.617.2)

## Data Availability

Data will be available upon request from the first or corresponding authors.

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
