# Peer review of "Effectiveness and Safety of Inactivated SARS-CoV-2 Vaccine (BBIBP-CorV) among Healthcare Workers: A Seven-Month Follow-Up Study at Fifteen Central Hospitals"

_vaccines, 2023, doi:10.3390/vaccines11050892_

Round 1

Reviewer 1 Report

It is an  observational study in fifteen hospitals on 1228 healthcare workers,in a follow-up period spanned nearly seven months and the aim is to evaluate the safety and effectiveness of the  Sinopharm vaccine.

With two distinct Google forms were employed are been collected outcomes that had arisen before the interview and any adverse events that occurred during the week following each vaccine dose.

The PCR-confirmed cases was higher in the unvaccinated than vaccinated groups as the number of workdays lost due to COVID-19 infection.

The adverse events were mild and they were local pain, fever, increase of the blood pressure.

The inactivated BBIBP-CorV vaccine (Sinopharm) was effective (effectiveness of 67% (95% CI 283 80-43%) calculated as a percentage of (1- adjusted HR), but it was not as effective against variants of SARS-CoV2.

The timeline study could ensure its effectiveness against the Delta variant.

Strengths: as reported by the authors, the study design (it was a multicenter study and it was used a multistage sampling technique) and the outcome analysis (with a subgroup analysis for the variables for outcomes).

Weaknesses: it is an observational study, but I think that also this type of study are usefel, more useful than case series for example.

Need improvements:

- the description of the sample should be inserted in materials and method.

- I don’t understand that “3.4. Literature review” in the results is appropriate. The “3.4. Literature review” should be inserted it in discussion to explain the results.

- Revise the results, please.

Author Response

Vaccines Journal, MDPI

Dear Editor

We include our revised manuscript title “Effectiveness and safety of inactivated SARS-CoV-2 vaccine (BBIBP-CorV) among healthcare workers: A seven-month follow-up study at fifteen Central hospitals” by Ashmawy et al., based on the professional and helpful reviewer’s comments (point-by-point) as following:

Reviewer 1

Comments and Suggestions for Authors

It is an observational study in fifteen hospitals on 1228 healthcare workers, in a follow-up period spanned nearly seven months and the aim is to evaluate the safety and effectiveness of the Sinopharm vaccine.

With two distinct Google forms were employed are been collected outcomes that had arisen before the interview and any adverse events that occurred during the week following each vaccine dose.

The PCR-confirmed cases was higher in the unvaccinated than vaccinated groups as the number of workdays lost due to COVID-19 infection.

The adverse events were mild and they were local pain, fever, increase of the blood pressure.

The inactivated BBIBP-CorV vaccine (Sinopharm) was effective (effectiveness of 67% (95% CI 283 80-43%) calculated as a percentage of (1- adjusted HR), but it was not as effective against variants of SARS-CoV2.

The timeline study could ensure its effectiveness against the Delta variant.

Strengths: as reported by the authors, the study design (it was a multicenter study and it was used a multistage sampling technique) and the outcome analysis (with subgroup analysis for the variables for outcomes).

Weaknesses: it is an observational study, but I think that also this type of study is usefel, and more useful than case series for example.

Need improvements:

- the description of the sample should be inserted in the materials and method.

Reply: Thank you for your systematic comment. We added section (2.2. Materials and methods) at line 95, to separate the sampling teqnichque from the study type and settings.

- I don’t understand that “3.4. Literature review” in the results is appropriate. The “3.4. Literature review” should be inserted it in discussion to explain the results.

Reply: Thank you for your suggestion. We added the literature review in the result section as these are our publication screening and analysis results. Then we discuss the results obtained as well as compare them with our findings in the discussion section.

- Revise the results, please.

Reply: Thank you, all results and their calculation revised again.

Reviewer 2 Report

Major concerns.

1. Please add the EC/IRB approval number to the "Study design and participants".

2. Was this study enrolled participants who had a history of SARS-CoV-2 infection? 
How to ensure that all participants never have had a previous SARS-CoV-2 infection? By RT-PCR, immunologic test or questionnaire/interview?

The previous infection may alter the outcome due to the immunity from infection plus vaccinations.

3. Lines 157-159, "the suspected case is any HCWs has three or more typical COVID-19 symptoms: fever, cough, general weakness, headache, myalgia, coryza, dyspnea, nausea/vomiting, diarrhea or altered mental status.".

Have you collected data focusing on anosmia and ageusia?
Both symptoms are likely specific to COVID-19.

4. Tables 1&2, the p-value came from the comparison between Unvaccinated and Vaccinated groups or included Total?

Minor concerns.

1. Table S9, "SBP/DBP after vaccination". Suggest re-classified following Table 4 "https://www.ahajournals.org/doi/10.1161/CIRCULATIONAHA.121.054602".
However, you may be using a further row for hypotension (<120/80 mmHg).

Comments.

1. Suggest using "SARS-CoV-2" instead of "SARS-CoV2" throughout the manuscript.

2. Line 141, suggests using "BBIBP-CorV" instead of the Sinopharm vaccine.

Author Response

Vaccines Journal, MDPI

Dear Editor

We include our revised manuscript title “Effectiveness and safety of inactivated SARS-CoV-2 vaccine (BBIBP-CorV) among healthcare workers: A seven-month follow-up study at fifteen Central hospitals” by Ashmawy et al., based on the professional and helpful reviewer’s comments (point-by-point) as following:

Reviewer 2

Comments and Suggestions for Authors

Major concerns.

  1. Please add the EC/IRB approval number to the "Study design and participants".

Reply: Thank you, approval number was added in line 105

  1. Was this study enrolled participants who had a history of SARS-CoV-2 infection?

How to ensure that all participants never have had a previous SARS-CoV-2 infection? By RT-PCR, immunologic test or questionnaire/interview?

The previous infection may alter the outcome due to the immunity from infection plus vaccinations.

Reply: Thank you very much for this important comment. We had the same concern while we were designing the study and tried to overcome this problem by considering the previous infection as a confounder and asking all the participants if they previously had a confirmed infection by the SARS-CoV-2. This was statistically non-significant between the 2 groups enrolled and the time of the infection, and if they were hospitalized or not, and we performed subgroup analysis by previous COVID-19, in addition to its being one of the adjusted factors in calculating vaccine effectiveness through (aHR).

  1. Lines 157-159, "the suspected case is any HCWs has three or more typical COVID-19 symptoms: fever, cough, general weakness, headache, myalgia, coryza, dyspnea, nausea/vomiting, diarrhea or altered mental status.".

Have you collected data focusing on anosmia and ageusia?

Both symptoms are likely specific to COVID-19.

Reply: Thank you for this comment. Yes, anosmia and ageusia were also typical symptoms of COVID-19 included in the Egyptian protocol and we were asking the participants for them besides other symptoms and if they have three of them, we considered the participant as a suspected case. We added them to line 161.

  1. Tables 1&2, the p-value came from the comparison between Unvaccinated and Vaccinated groups or included Total?

Reply: Test of significance used between the 2 groups: Chi-square, Fischer Exact, and Mann-Whitney. This sentence was added in the footnotes of the 2 tables.

Minor concerns.

  1. Table S9, "SBP/DBP after vaccination". Suggest re-classified following Table 4 "https://www.ahajournals.org/doi/10.1161/CIRCULATIONAHA.121.054602".

However, you may be using a further row for hypotension (<120/80 mmHg).

Reply: Thank you for this suggestion. Recalculation was performed and data changed according to ACC/AHA classification in Table S9.

Comments.

  1. Suggest using "SARS-CoV-2" instead of "SARS-CoV2" throughout the manuscript.

Reply: corrected throughout the entire MS.

  1. Line 141, suggests using "BBIBP-CorV" instead of the Sinopharm vaccine.

Reply: corrected throughout the entire MS.

Round 2

Reviewer 2 Report

After revision, this manuscript is cleared and acceptable for published.